# Closed-loop optogenetic activation of peripheral or central neurons modulates feeding in freely moving *Drosophila*

**Pierre-Yves Musso[1], Pierre Junca[1], Meghan Jelen[1], Damian Feldman-Kiss[1], Han Zhang[2], Rachel CW Chan[2†], Michael D Gordon[1]\***

[1]Department of Zoology, Life Sciences Institute, University of British Columbia, Vancouver, Canada; [2]Department of Physics and Astronomy, University of British Columbia, Vancouver, Canada

**Abstract** Manipulating feeding circuits in freely moving animals is challenging, in part because the timing of sensory inputs is affected by the animal's behavior. To address this challenge in *Drosophila,* we developed the Sip-Triggered Optogenetic Behavior Enclosure ('STROBE'). The STROBE is a closed-looped system for real-time optogenetic activation of feeding flies, designed to evoke neural excitation coincident with food contact. We previously demonstrated the STROBE's utility in probing the valence of fly sensory neurons (Jaeger et al., 2018). Here we provide a thorough characterization of the STROBE system, demonstrate that STROBE-driven behavior is modified by hunger and the presence of taste ligands, and find that mushroom body dopaminergic input neurons and their respective post-synaptic partners drive opposing feeding behaviors following activation. Together, these results establish the STROBE as a new tool for dissecting fly feeding circuits and suggest a role for mushroom body circuits in processing naïve taste responses.
DOI: https://doi.org/10.7554/eLife.45636.001

**\*For correspondence:**
gordon@zoology.ubc.ca

**Present address:** [†]Department of Computer Science, University of Toronto, Toronto, Canada

**Competing interests:** The authors declare that no competing interests exist.

## Introduction

*Drosophila melanogaster* has emerged as a leading model for understanding sensory processing related to food approach, avoidance, and consumption behaviors. However, although the gustatory system is recognized as mediating a critical final checkpoint in determining food suitability, much remains to be learned about the neural circuits that process taste information in the fly brain.

Like mammals, flies detect several taste modalities, each of which promotes food acceptance or rejection (*Liman et al., 2014*; *Marella et al., 2006*; *Yarmolinsky et al., 2009*). Taste compounds activate gustatory receptor neurons (GRNs) localized on the fly's proboscis, legs, wings, and ovipositor (*Scott, 2018*). Among the different classes of GRNs present, cells expressing the Gustatory Receptor Gr64f respond to sweet compounds and induce strong acceptance behavior. Conversely, GRNs labeled by Gr66a respond to bitter compounds and evoke avoidance (*Dahanukar et al., 2001*; *Dahanukar et al., 2007*; *Jiao et al., 2008*; *Kwon et al., 2014*; *Kwon et al., 2011*; *Marella et al., 2006*; *Thorne et al., 2004*; *Wang et al., 2004*). GRNs connect directly or indirectly to the subesophageal zone (SEZ) of the fly brain (*Ito et al., 2014*; *Rajashekhar and Singh, 1994*; *Scott, 2018*; *Stocker and Schorderet, 1981*). Taste processing in the SEZ involves local modulatory interneurons (*Chu et al., 2014*; *Pool et al., 2014*), second-order neurons projecting locally or to other brain regions (*Kain and Dahanukar, 2015*; *Kim et al., 2017*; *Yapici et al., 2016*), motor neurons driving feeding subprograms (*Gordon and Scott, 2009*; *Hampel et al., 2011*; *Manzo et al., 2012*; *Rajashekhar and Singh, 1994*), and command neurons driving the complete feeding program (*Flood et al., 2013*).

Taste processing is not only involved in acute feeding events, but also in the formation of associative memories, which are aversive following exposure to bitter taste (*Masek et al., 2015*; *Kirkhart and Scott, 2015*) or positive following sugar consumption (*Tempel et al., 1983*). Memory formation occurs mainly in a central brain structure called the mushroom body (MB), composed of ~2000 Kenyon cells per hemisphere (*Heisenberg et al., 1985*). The MBs receive sensory information that is assigned a positive or negative output valence via coincident input from dopaminergic neurons (DANs) (*Perisse et al., 2013*; *Waddell, 2010*). Little is known about how taste information is relayed to the MBs, but taste projection neurons (TPNs) connected to bitter GRNs indirectly drive activation of the paired posterior lateral cluster 1 (PPL1) DANs (*Kim et al., 2017*). PPL1 neurons signal punishment to MBs and are required for aversive taste memory formation (*Aso et al., 2012*; *Aso et al., 2010*; *Claridge-Chang et al., 2009*; *Kim et al., 2017*; *Kirkhart and Scott, 2015*; *Masek et al., 2015*). Conversely, the protocerebrum anterior medial (PAM) cluster of DANs signals rewarding information and is involved in the formation of appetitive memories (*Burke et al., 2012*; *Huetteroth et al., 2015*; *Liu et al., 2012*; *Yamagata et al., 2015*). Although they have well-established roles in memory formation, PPL1 and PAM involvement in feeding has not been extensively investigated.

Kenyon cells and DANs make connections to specific mushroom body output neurons (MBONs) within discrete compartments of the MBs. MBONs project to protocerebral integration centers and are required for memory formation and retrieval (*Aso et al., 2014a*; *Aso et al., 2014b*; *Aso et al., 2014b*; *Bouzaiane et al., 2015*; *Felsenberg et al., 2017*; *Ichinose et al., 2015*; *Masek et al., 2015*; *Owald et al., 2015*; *Perisse et al., 2016*; *Plaçais et al., 2013*; *Séjourné et al., 2011*; *Takemura et al., 2017*; *Tanaka et al., 2008*). An emerging model is that DAN/MBON pairs innervating a specific MB compartment produce behavioral responses of opposing valence, and that KC-MBON synapses in that compartment are depressed upon DAN activation (*Cohn et al., 2015*; *Felsenberg et al., 2017*; *Perisse et al., 2016*; *Séjourné et al., 2011*; *Takemura et al., 2017*). While MBONs are known to modulate innate behaviors such as taste sensitivity (*Masek et al., 2015*) and food seeking behavior (*Tsao et al., 2018*), the possible contribution of MB input and output circuits to feeding behavior remains unclear.

Manipulating neural activity is a powerful method for assessing neural circuit function. Silencing neuron populations in freely behaving flies, which forces the neurons into a chronic 'off' state to mimic a situation where the fly never encounters an activating stimulus, is a straightforward way to determine their necessity in feeding. (*Fischler et al., 2007*; *Gordon and Scott, 2009*; *LeDue et al., 2015*; *LeDue et al., 2016*; *Mann et al., 2013*; *Marella et al., 2012*; *Pool et al., 2014*). However, gain-of-function experiments for feeding and taste, or any other actively sensed stimulus, are more complicated. Behaviors produced by forcing a neuron into a stimulus- and behavior-independent 'on' state can be difficult to interpret. The possible exception is activation of a neuron that elicits a stereotyped motor program, but even these situations are more easily interpreted in a tethered fly where the effect of a single activation can be monitored (*Chen and Dahanukar, 2017*; *Flood et al., 2013*; *Gordon and Scott, 2009*; *Marella et al., 2006*; *Masek et al., 2015*). To effectively probe the sufficiency of neuron activation during feeding events, it would be ideal to temporally couple activation with feeding.

Recently, three new systems for closed-loop optogenetic control of feeding flies have emerged: the optoPAD and STROBE, developed as additions to the FlyPAD system; and the optoFLIC, built on the FLIC platform (*Jaeger et al., 2018*; *May et al., 2019*; *Moreira et al., 2019*; *Ro et al., 2014*; *Steck et al., 2018*). Here, we provide a more extensive characterization of the STROBE and its utility. We demonstrate that coincident activation of sweet GRNs with feeding on agar drives appetitive behavior, and bitter GRN activation elicits aversion. These effects are modulated by starvation and can be inhibited by the presence of chemical taste ligands of the same modality. We also show that activation of central feeding circuit neurons produces repetitive, uncontrolled food interactions, demonstrating the STROBE's efficacy in manipulating both peripheral and central neurons. We then establish that activation of PPL1 neurons negatively impacts feeding, while activating PAM neurons promotes it. Finally, in agreement with mushroom body circuit models, activating MBONs drives feeding responses in opposition to the DANs from the same MB compartment.

## Results

### The STROBE triggers light activation temporally coupled with food interactions

The FlyPAD produces capacitance signals that reflect a fly's interaction with food in either of two sensors (or 'channels') in a small arena (*Figure 1A*, *Figure 1—figure supplement 1A*) (*Itskov et al., 2014*). When a fly physically bridges the two electrodes of a sensor by standing on one electrode and making contact with food sitting on the other, it produces a rise in capacitance. This signal, which is acquired at 100 Hz, is then decoded *post hoc* by an algorithm designed to identify sipping events. We designed the STROBE to track the raw capacitance signal in real-time and trigger lighting within the arena during sips (*Figure 1B*). To achieve this, we built arena attachments that consist of a lighting PCB carrying two LEDs of desired colors positioned above the channels of a FlyPAD arena. Each PCB is surrounded by a lightproof housing to isolate the arenas from other light sources (*Figure 1—figure supplement 1B–C*).

In order to trigger optical stimulation with short latency upon sip initiation, we designed an algorithm that applies a running minima filter to the capacitance signal to detect when a fly is feeding. If the capacitance surpasses a set threshold above the minimum value recorded within the preceding 10 cycles (100 ms), the LED is turned on and remains illuminated until the threshold is no longer exceeded. By definition, this means that elevated plateaus of capacitance that last longer than 100 ms will produce a lighting response only within the first 100 ms. Because this algorithm is run on a field-programmable gate array (FPGA), the signal to lighting transition times are theoretically on the order of tens of milliseconds, providing a rapid response following the initiation of a sip. We confirmed the short latency of activation with a wire standing in for the fly's proboscis. Based on video captured at 178 frames per second, the latency of LED activation following a touch was 37 ± 17 ms (*Figure 1—figure supplement 2*). This latency is short enough to ensure LED triggering during sips, which generally last longer than 100 ms (*Itskov et al., 2014*). Indeed, video of flies feeding in the STROBE, shot at 60 frames per second, consistently showed light activation in the same frame as the fly's proboscis fully extending onto the food (*Figure 1—figure supplement 2* and *Video 1*).

The STROBE records the state of the lighting activation system (on/off) and transmits this information through USB to the PC, where it is received and interpreted by a custom end-user program. This program displays the capacitance signals from each fly arena in real-time, as well as its lighting state. It also counts the number of LED activations over the course of the experiment (*Figure 1C*; *Figure 1—figure supplement 1D–F*). To confirm that the STROBE algorithm triggers the LED during sips detected by the original *post hoc* FlyPAD algorithm, we first used both algorithms to analyze the capacitance signal from a short (~11 s) feeding bout (*Figure 1D*). Visually, this showed that each time a sip is detected with the FlyPAD algorithm, the STROBE algorithm triggers the LED at a similar time. However, we also noted that the STROBE algorithm triggers more LED activations than the number of sips called by the FlyPAD algorithm. We confirmed these observations on a larger scale by examining the correlation between the output of each algorithm in 1 min bins across a full 1 hr experiment (*Figure 1E*). Here, there is a strong correlation between the two ($R^2$ = 0.963), with the number of LED illuminations triggered by the STROBE algorithm being about 1.4 times the number of sips detected by the FlyPAD algorithm. This increased number is likely the consequence of the FlyPAD algorithm filtering out capacitance changes not adhering to certain criteria of shape and duration (*Itskov et al., 2014*). Since these parameters are, by definition, unknown at sip onset, the STROBE cannot use them as criteria. Thus, we expect that a fraction of LED activations in the STROBE are actually triggered by more fleeting interactions with the food. Indeed, video of flies in the STROBE confirmed that a subset of leg touches triggered light activation (*Video 1*). Thus, we defined each LED activation as representing a food 'interaction', the majority of which are sips. Since flies detect tastes on multiple body parts, including the legs, even non-sip interactions are likely still relevant to taste processing and feeding initiation.

### Activation of GRNs modifies feeding behavior

To validate the utility of the STROBE, we first tested flies expressing CsChrimson, a red-light activated channel, in either sweet or bitter GRNs (*Klapoetke et al., 2014*). Flies were given the choice between two identical neutral food options (plain agar), one of which triggered light activation.

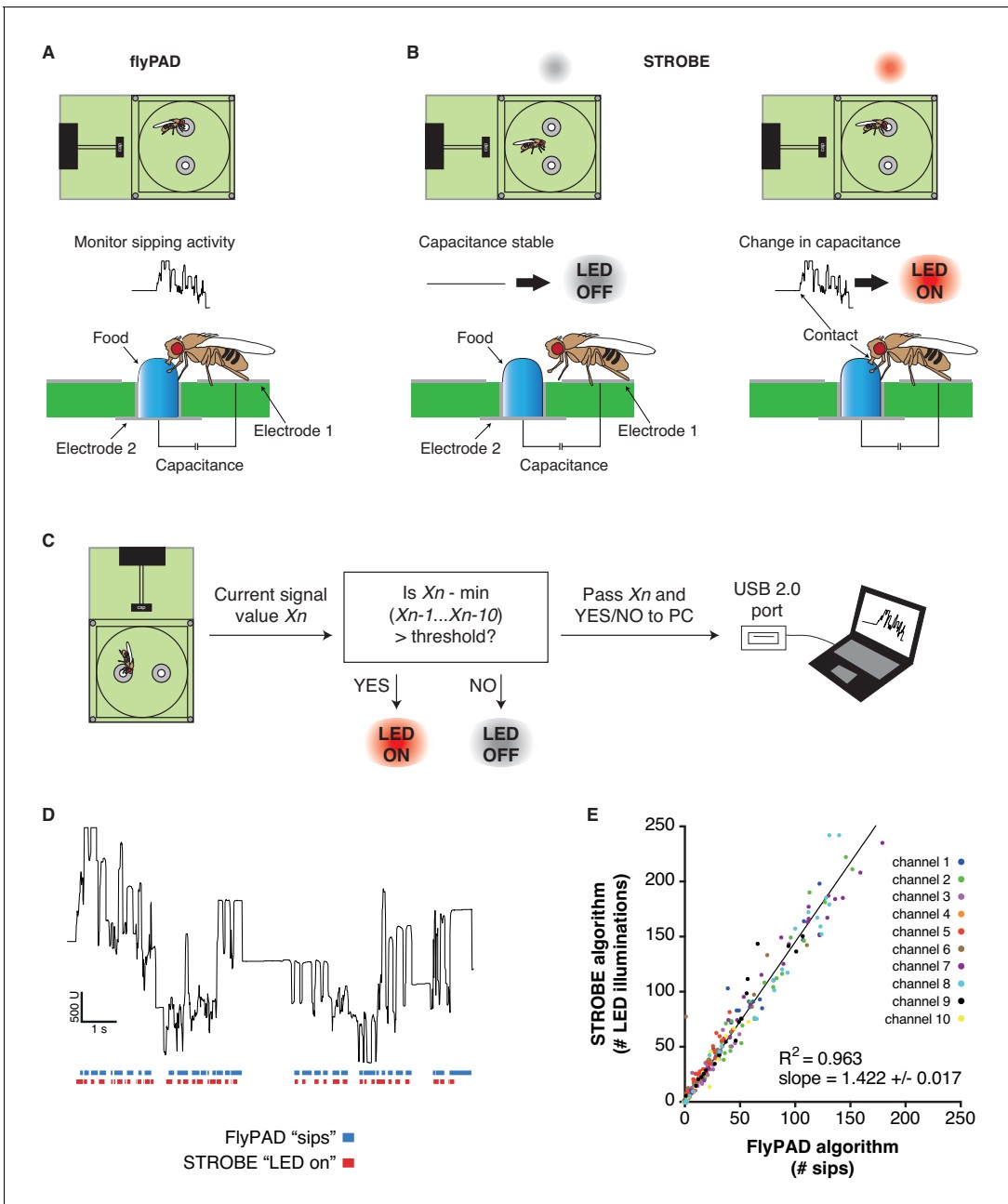

**Figure 1.** The STROBE setup. (**A**) Concept of the FlyPAD: The interaction between the fly's proboscis and the food is detected as a change in capacitance between two electrodes: electrode 1, on which the fly stands, and electrode 2, on which the food is placed. (**B**) Concept of the STROBE: when the fly is not interacting with the food, no change of capacitance is detected and the LED is OFF (left); when the fly sips, changes in capacitance turn the LED ON (right). (**C**) Flowchart of the STROBE signal processing algorithm. (**D**) Example of capacitance changes during a feeding bout, and the associated sips called by the FlyPAD (blue) and STROBE lighting events (red). (**E**) Comparison of the sip numbers called by the FlyPAD algorithm and LED illuminations triggered by STROBE algorithm. Sips/illuminations were counted in 1 min bins across a 1 hr experiment for 10 different channels (five arenas). Bins with neither sips detected by the FlyPAD algorithm nor illuminations triggered by the STROBE algorithm were excluded from analysis, as these were deemed times when the fly was not interacting with the food.

DOI: https://doi.org/10.7554/eLife.45636.002

The following source data and figure supplements are available for figure 1:

**Source data 1.** This file contains all the raw numerical data for *Figure 1* and its associated figure supplements.
DOI: https://doi.org/10.7554/eLife.45636.005
**Figure supplement 1.** The STROBE setup.
DOI: https://doi.org/10.7554/eLife.45636.003

*Figure 1 continued on next page*

*Figure 1 continued*

**Figure supplement 2.** LED illumination is triggered with short latency following touch.
DOI: https://doi.org/10.7554/eLife.45636.004

Under these conditions, flies expressing functional CsChrimson in sweet neurons under the control of *Gr64f-GAL4* showed a dramatic preference towards feeding on the light-triggering food, while control flies of the same genotype that were not pre-fed all-*trans*-retinal, and thus carried non-functional CsChrimson, displayed no preference (*Figure 2A–E*). The number of interactions on the light-triggering side of the chamber was dependent on light intensity, with increasing interactions up 6.5 mW/cm$^2$, above which interaction numbers decreased as a function of intensity. However, preference was relatively stable above a threshold of 1.85 mW/cm$^2$ (*Figure 2D–E*). Additionally, interactions generally accumulated linearly over time at all intensities, with a stable preference index established within the first 15 min of a one-hour experiment (*Figure 2—figure supplement 1*). This suggests that neither sensitization nor adaptation occurs during the course of the experiment. Moreover, the flies do not appear to adjust their behavior in response to the perceived mismatch between the sweet taste and lack of energy content of the food source.

As expected, flies expressing functional CsChrimson in bitter sensing neurons under the control of *Gr66a-GAL4* strongly avoided neuronal activation in the STROBE by engaging in fewer interactions with the light-triggering food source (*Figure 2F–J*). Once again the behavioral response was intensity-dependent, with maximum suppression of interactions occurring at the highest intensity tested (16.4 mW/cm$^2$). As with sweet GRN activation, the flies' preference develops in roughly the first 15 min and remains relatively stable for the rest of the hour (*Figure 2—figure supplement 2*). Since the light intensities eliciting maximal effect for Gr64f and the Gr66a activation are 6.5 mW/cm$^2$ and 16.4 mW/cm$^2$, respectively, we decided to use the intermediate value of 11.2 mW/cm$^2$ as the intensity for all further experiments. Using a full set of genotypic and non-retinal controls, we confirmed that these conditions produced robust and specific preference behaviors for both sweet and bitter GRN activation (*Figure 2—figure supplement 3*).

Next, we sought to test whether GRN activation affects feeding per se, rather than simply driving non-ingestive food interactions. We repeated sweet and bitter GRN activation in the STROBE with the addition of blue dye to one food source and red dye to the other. We then calculated a *post hoc* preference based on the number of individual flies showing blue versus red dye in their abdomen following the assay, and compared this to the preference calculated based on food interactions measured by the STROBE. In all cases, these studies confirmed that the changes in food interactions driven by GRN activation are accompanied by strong effects on food ingestion in the expected direction (*Figure 2—figure supplement 4*).

## Starvation modulates the behavioral impact of GRN activation

Starvation duration has a well-known impact on fly feeding – the longer flies are food deprived, the more they will initiate and sustain feeding on sweet foods (*Dus et al., 2013*; *Dus et al., 2011*; *Inagaki et al., 2012*; *Inagaki et al., 2014*; *Scheiner et al., 2004*). To determine if similar effects would manifest in the STROBE, we tested GRN activation after different periods of food deprivation (*Figure 3A,B*). Consistent with its effect on sugar feeding, starvation increased flies' preference for the light-triggering food in the STROBE when their sweet neurons expressed functional CsChrimson (*Figure 3C*). This elevated preference index is driven by a dramatic increase in interaction number

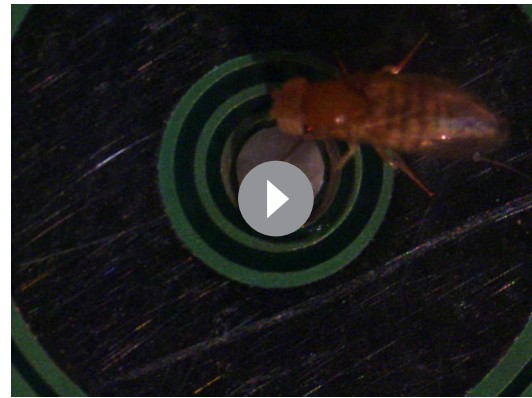

**Video 1.** This movie shows an individual fly feeding on the light-triggering food in a STROBE arena.
DOI: https://doi.org/10.7554/eLife.45636.006

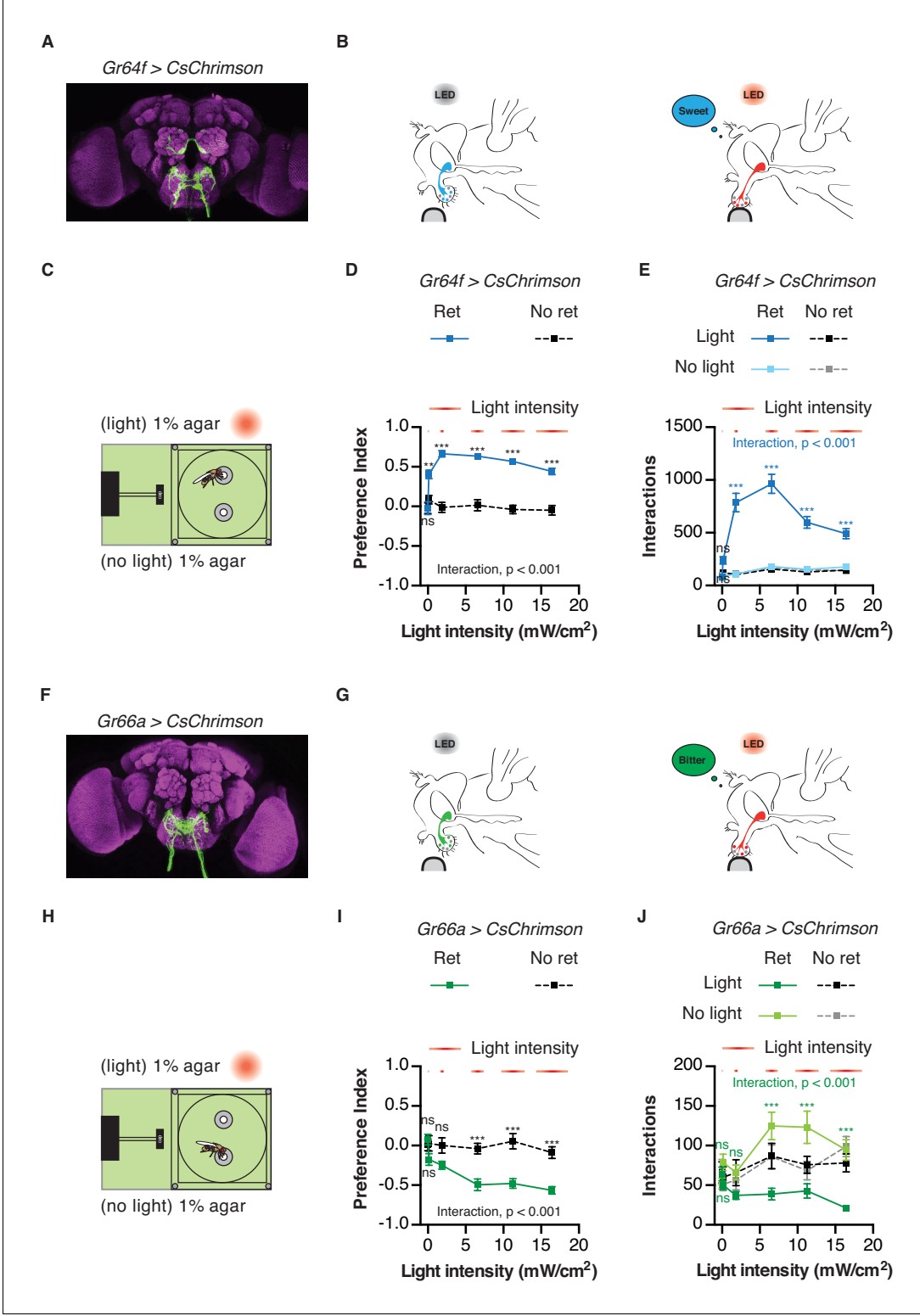

**Figure 2.** Activation of sweet and bitter sensory neurons drives feeding preferences in the STROBE. (**A**) Immunofluorescent detection of *UAS-CsChrimson.mVenus* driven by *Gr64f-GAL4*. (**B**) Schematic illustrating STROBE activation of sweet neurons. (**C**) Experimental setup: both channels are filled with 1% agar, only one is paired to LED activation. (**D–E**) Relationship between light intensity and light side preference (**D**) or interaction numbers (**E**) for *Gr64f > CsChrimson* flies pre-fed retinal (blue squares) or not fed retinal (black squares). (**F**) Expression of *UAS-CsChrimson.mVenus* driven by *Gr66a-GAL4*. (**G**) Schematic illustrating STROBE activation of
*Figure 2 continued on next page*

*Figure 2 continued*

bitter neurons. (H) Experimental setup: both channels contain plain 1% agar. (I–J) Relationship between light intensity and light side preference (I) or interaction numbers (J) *Gr66a > CsChrimson* flies pre-fed retinal (green squares) or not fed retinal (black squares). Values represent mean ± SEM. n = 30–37 (D–E) or 19–28 (H–I). Statistical tests: two-way ANOVA and Bonferroni post hoc: ns p>0.05, **p<0.01, ***p<0.001. Colored asterisks represent significance between sips on each side for the retinal group.

DOI: https://doi.org/10.7554/eLife.45636.008

The following source data and figure supplements are available for figure 2:

**Source data 1.** This file contains all the raw numerical data for *Figure 2* and its associated figure supplements.
DOI: https://doi.org/10.7554/eLife.45636.013
**Figure supplement 1.** Behavioral dynamics during sweet GRN stimulation.
DOI: https://doi.org/10.7554/eLife.45636.009
**Figure supplement 2.** Behavioral dynamics during bitter GRN stimulation.
DOI: https://doi.org/10.7554/eLife.45636.010
**Figure supplement 3.** Genetic controls for Gr66a and Gr64f do not show any preference in the STROBE.
DOI: https://doi.org/10.7554/eLife.45636.011
**Figure supplement 4.** GRN activation in the STROBE drives ingestion behavior.
DOI: https://doi.org/10.7554/eLife.45636.012

(*Figure 3D*). In contrast to its impact on sweet sensory neurons, starvation had no significant effect on light avoidance mediated by bitter neuron activation under these conditions (*Figure 3E,F*). Starvation also had little to no effect on the timing of food choice throughout the assay, with all groups establishing their peak preference within the first 10–15 min and largely maintaining it throughout the assay (*Figure 3—figure supplements 1* and *2*).

## Chemical taste ligands suppress the impact of light-induced attraction and avoidance

We next asked whether the presence of sweet or bitter ligands would interfere with light-driven behavior in the STROBE. For example, if sugar is placed in both food options, will this reduce the salience of sweet GRN activation by light? Indeed, adding increasing concentrations of sucrose to both food options caused dose-dependent inhibition of *GR64f > CsChrimson* flies' preference for the light-triggering food (*Figure 4A,B*). This change is driven by a progressively higher number of interactions on the no-light side, with relatively constant interaction numbers on the light side as sucrose concentration increases (*Figure 4—figure supplement 1A,B*). On the other hand, the addition of sucrose mildly enhanced the negative preferences driven by STROBE activation of Gr66a bitter neurons (*Figure 4C*). This effect appears to manifest from the increasing attractiveness of the no-light side coupled with unwavering and near total avoidance of the light-triggering side (*Figure 4—figure supplement 1C*). The same pattern is mirrored by the addition of the bitter compound denatonium to both sides: dose-dependent inhibition of the aversion shown by *Gr66a > CsChrimson* flies (*Figure 4D,E*; *Figure 4—figure supplement 1D,E*), and little to no effect on the attraction of *Gr64f > CsChrimson* flies to the light side (*Figure 4F*; *Figure 4—figure supplement 1F*). In general, as with other experiments, we observed no substantial change in preference after establishment in the first 10–15 min (*Figure 4—figure supplements 2–5*). This suggests that flies are not displaying satiety effects from sugar ingestion or taste-independent effects from the consumption of denatonium.

## Activation of the 'feeding-neuron' drives extreme sipping behavior

Can the STROBE affect feeding behavior through the activation of central neurons, in addition to those in the periphery? Although the precise nature of higher-order taste circuits is still unclear, several neurons have been identified in the SEZ that influence feeding behavior (*Chu et al., 2014*; *Inagaki et al., 2014*; *Inagaki et al., 2012*; *Jourjine et al., 2016*; *Kain and Dahanukar, 2015*; *LeDue et al., 2016*; *Marella et al., 2012*; *Pool et al., 2014*; *Yapici et al., 2016*). One of them, the 'feeding-neuron' (Fdg), acts as a command neuron for the proboscis extension response, and shows activity in response to food stimulation only following starvation (*Flood et al., 2013*). Strikingly, Fdg activation in the STROBE produced an extremely high number of interactions on the light-triggering

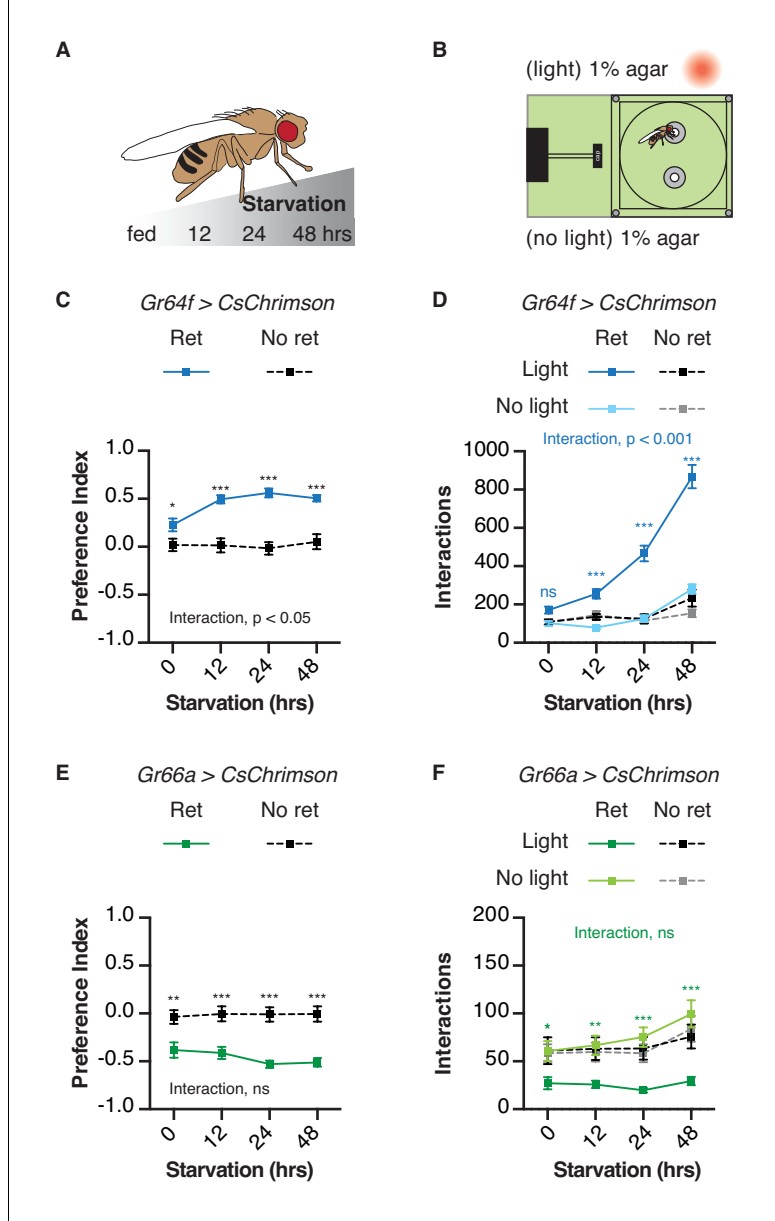

**Figure 3.** Behavioral impact of GRN activation is modulated by starvation. (**A**) Protocol: flies are subjected to increasing period of starvation (12 hr, 24 hr, 48 hr) prior to the STROBE experiment. (**B**) Experimental setup: both channels contain plain 1% agar. (**C–D**) The effect of starvation on light side preference (**C**) and food interaction numbers (**D**) of flies expressing CsChrimson in sweet neurons. (**E–F**) The effect of starvation on preference for the light side (**E**) and interaction numbers (**F**) of flies expressing CsChrimson in bitter neurons. Values represent mean ± SEM. n = 21–30. Statistical tests: two-way ANOVA with Bonferroni post hoc. ns p>0.05, *p<0.05, **p<0.01, ***p<0.001. Colored asterisks represent significance between sips on each side for the retinal group.
DOI: https://doi.org/10.7554/eLife.45636.014

The following source data and figure supplements are available for figure 3:

**Source data 1.** This file contains all the raw numerical data for *Figure 3* and its associated figure supplements.
DOI: https://doi.org/10.7554/eLife.45636.017
**Figure supplement 1.** Behavioral dynamics during sweet GRN stimulation following different starvation times.
DOI: https://doi.org/10.7554/eLife.45636.015
**Figure supplement 2.** Behavioral dynamics during bitter GRN stimulation following different starvation times.
DOI: https://doi.org/10.7554/eLife.45636.016

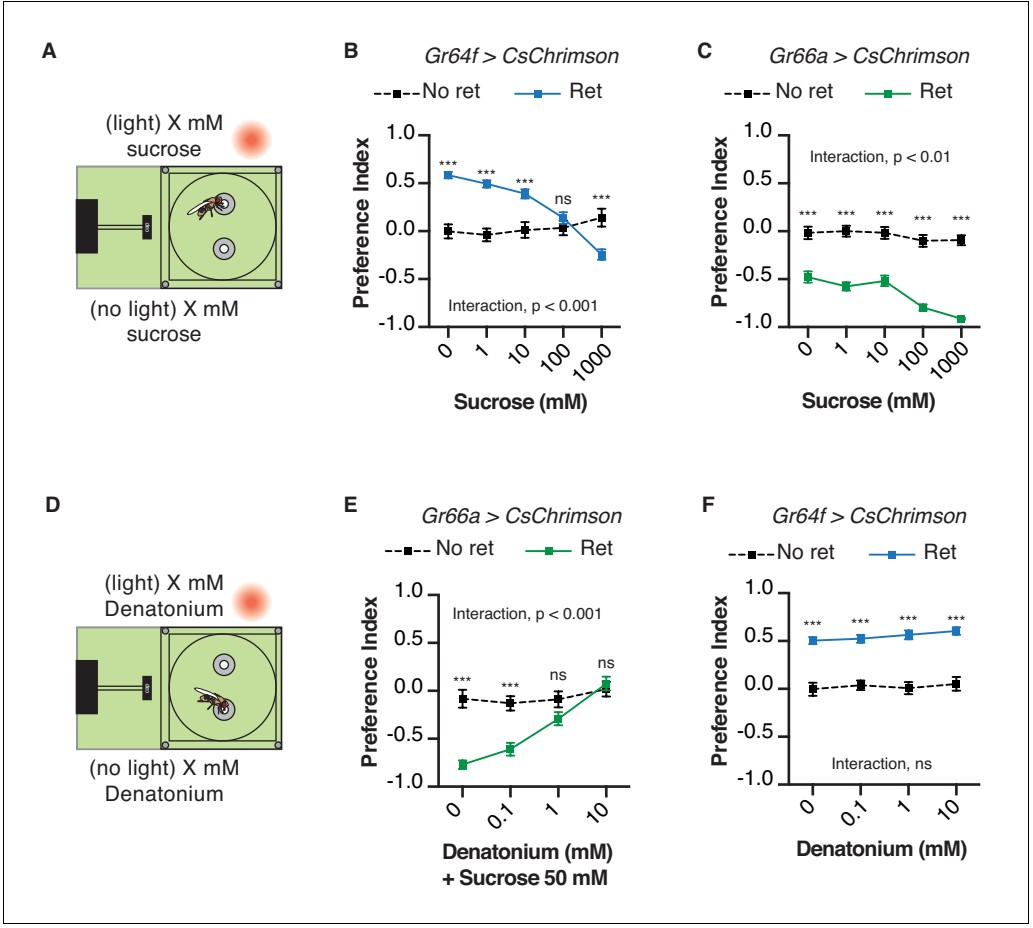

**Figure 4.** Chemical taste ligands suppress impact of light evoked GRN activity. (**A**) Experimental setup: both channels contain the same sucrose concentration (1, 10, 100, or 1000 mM) in 1% agar. (**B–C**) The effect of sucrose concentration on the light side preference of *Gr64f > CsChrimson* (**B**) or *Gr66a > CsChrimson* (**C**). (**D**) Experimental setup: both channels contain the same denatonium concentration (0, 0.1, 1, or 10 mM). For *Gr66a > CsChrimson* activation, both channels also contain 50 mM sucrose. (**E–F**) The effect of denatonium concentration on the light side preference of *Gr66a > CsChrimson* (**E**) or *Gr64f > CsChrimson* (**F**). Values represent mean ± SEM. n = 25–51. Statistical tests: two-way ANOVA with Bonferroni post hoc. ns p>0.05; ***p<0.001.
DOI: https://doi.org/10.7554/eLife.45636.018

The following source data and figure supplements are available for figure 4:

**Source data 1.** This file contains all the raw numerical data for *Figure 4* and its associated figure supplements.
DOI: https://doi.org/10.7554/eLife.45636.024
**Figure supplement 1.** Chemical taste ligands suppress impact of light evoked GRN activity.
DOI: https://doi.org/10.7554/eLife.45636.019
**Figure supplement 2.** Behavioral dynamics during sweet GRN stimulation with sugar-containing food.
DOI: https://doi.org/10.7554/eLife.45636.020
**Figure supplement 3.** Behavioral dynamics during bitter GRN stimulation with sugar-containing food.
DOI: https://doi.org/10.7554/eLife.45636.021
**Figure supplement 4.** Behavioral dynamics during bitter GRN stimulation with bitter-containing food.
DOI: https://doi.org/10.7554/eLife.45636.022
**Figure supplement 5.** Behavioral dynamics during sweet GRN stimulation with bitter-containing food.
DOI: https://doi.org/10.7554/eLife.45636.023

food, resulting in a nearly complete preference for that side (*Figure 5*; *Video 2*). Thus, the STROBE can effectively modulate feeding behavior via the activation of either peripheral or central neurons.

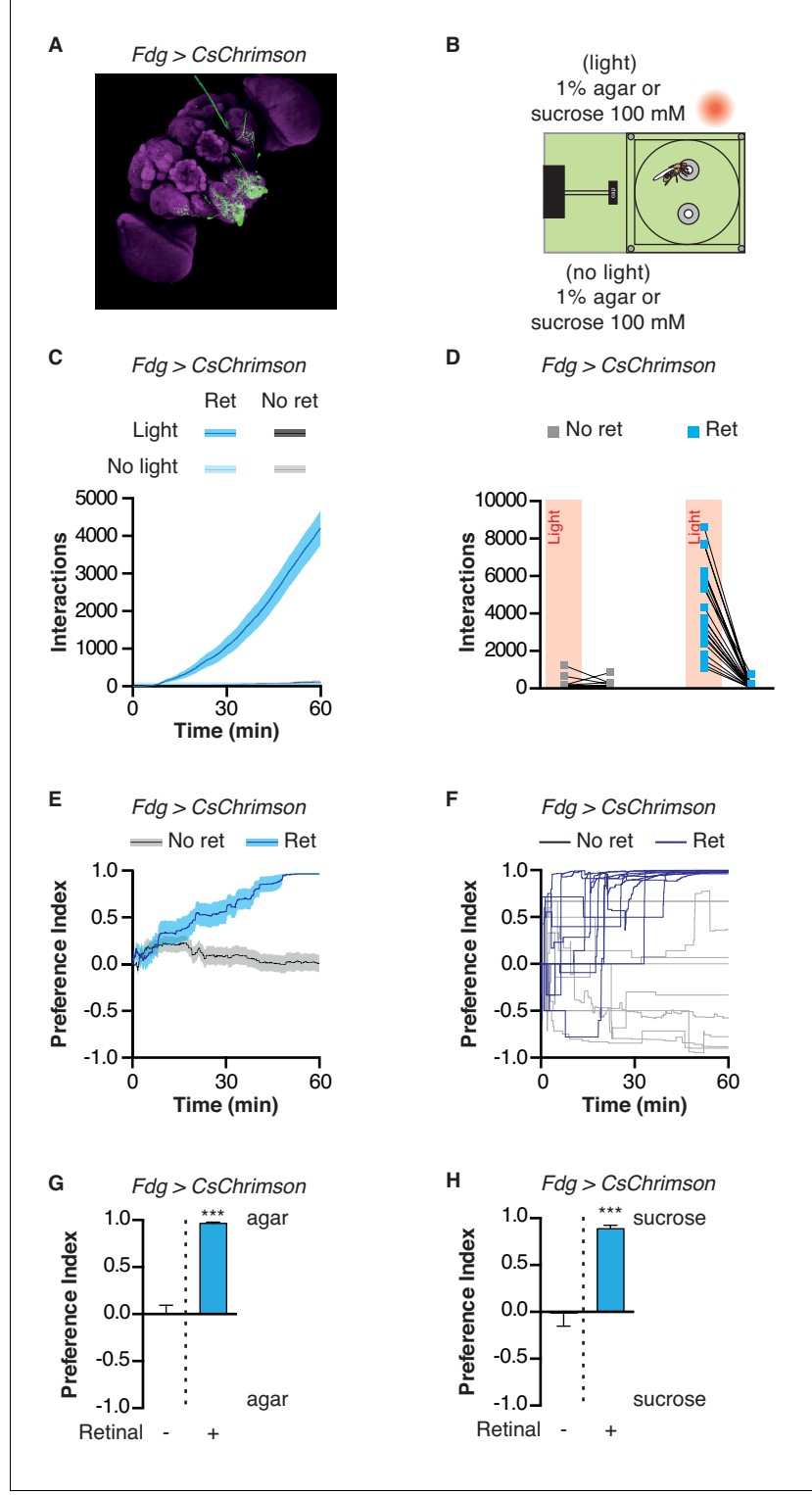

**Figure 5.** Activation of feeding command neurons elicits extreme sipping behavior. (**A**) Immunofluorescent detection of *Fdg > CsChrimson*. (**B**) Experimental setup: both channels contain either 1% agar or 100 mM sucrose. (**C**) Cumulative food interaction numbers for the population of *Fdg > Chrimson* flies over the course of a 1 hr experiment. (**D**) Total interaction numbers for individual flies. (**E–F**) Preference index for *Fdg > Chrimson* flies over the course of a 1 hr experiment averaged (**E**) or for ten individual flies (**F**). (**G–H**) Preference index for *Fdg > Chrimson* flies after one hour spent in the STROBE with agar (**G**) or 100 mM sucrose (**H**) . Values are mean ± SEM. n = 14–33. Statistical tests: *t*-test. ***p<0.001.

*Figure 5 continued on next page*

*Figure 5 continued*

DOI: https://doi.org/10.7554/eLife.45636.025

The following source data is available for figure 5:

**Source data 1.** This file contains all the raw numerical data for *Figure 5*.

DOI: https://doi.org/10.7554/eLife.45636.026

## Manipulating mushroom body extrinsic neurons modifies feeding behavior

Given the important role of the MB in assigning valence to stimuli during learning, we next asked whether MB circuits acutely impact attractive and aversive feeding responses. PPL1 DANs signal punishment or aversive information to the MBs, and thus their activation in the STROBE is predicted to drive avoidance of the light-triggering food (*Figure 6A*) (*Aso et al., 2012*; *Aso et al., 2010*; *Das et al., 2014*; *Kirkhart and Scott, 2015*; *Masek et al., 2015*). Flies expressing functional CsChrimson in the α3/α'3 subset of PPL1 dopaminergic neurons (*MB308B-GAL4*) showed a negative preference towards light when 100 mM sucrose was present in both options, but not if the food was plain agar (*Figure 6C*; *Figure 6—figure supplement 1A–C*). Performing the same experiment with activation of a broader subset of PPL1 neurons (*MB065B-GAL4*) led to stronger avoidance in the presence of 100 mM sucrose or plain agar (*Figure 6—figure supplement 1A–C*). Interestingly, activation of MBONs post-synaptic to PPL1 (*MB093C-GAL4* and *MB026B-GAL4*) produced strong attraction in either context (*Figure 6D*; *Figure 6—figure supplement 1A–C*).

The PAM cluster of DANs is generally thought to signal appetitive reward to the MBs (*Burke et al., 2012*; *Huetteroth et al., 2015*; *Lin et al., 2014*; *Liu et al., 2012*; *Yamagata et al., 2015*). Following the same principle described above, PAM activation should drive appetitive behavior, while stimulation of MBONs within the same compartment is predicted to elicit aversion (*Figure 6E*). Indeed, activating the β2m,β'2 p PAM subset (*MB056B-GAL4* and *MB301B-GAL4*) in the STROBE led to attraction (*Figure 6F,G*; *Figure 6—figure supplement 2*). On the other hand, activating the corresponding β2m,β'2 p,γ5 MBONs (*MB011B-GAL4* and *MB210B-GAL4*) produced avoidance when sucrose was present (*Figure 6H*, *Figure 6—figure supplement 2*).

Another subset of PAMs, targeting the γ3 compartment, was recently shown to encode a negative valence and induce appetitive memories following transient inhibition by the satiety peptide Allatostatin-A (*Yamagata et al., 2016*). Consistent with these results, activation of PAM γ3 neurons (*MB441B-GAL4* and *MB195B-GAL4*) in the STROBE produced light avoidance (*Figure 6I*; *Figure 6—figure supplement 3*), while activating the corresponding β'1,γ3 MBONs (*MB083C-GAL4* and *MB110C-GAL4*) was attractive (*Figure 6J*; *Figure 6—figure supplement 3*). Thus, PAMs targeting different MB compartments can be either attractive or aversive in the context of feeding.

Finally, we chose the PPL1 cluster and their corresponding MBONs to examine the impact of MB circuits on feeding in greater detail. First, we asked whether the interaction preferences driven by activating these populations reflected genuine changes in feeding behavior. Strikingly, dye ingestion in the STROBE consistently produced stronger measures of preference than those calculated with interaction numbers, mirroring the effects seen with GRN activation (*Figure 7*). This was true across a range of stimulus intensities, even when the differences in interaction numbers were quite low. Interestingly, in contrast to sweet GRN stimulation, which elicited peak interaction numbers midway in the series of intensities tested, *MB026B-GAL4* MBON activation produced linearly increasing interactions across the range of intensities (*Figure 7H*).

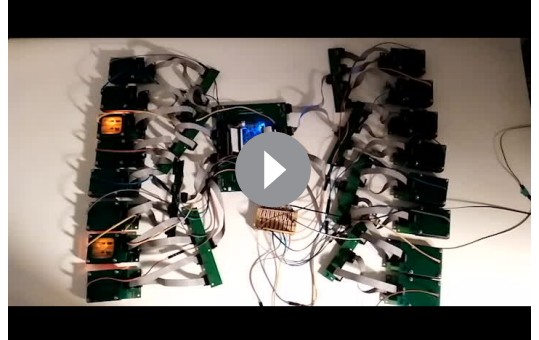

**Video 2.** This movie shows the full STROBE system. Each chamber contains a fly expressing CsChrimson under the control of *Fdg-GAL4*. The flies on the left side have been fed all-*trans* retinal. The flies on the right have not been fed retinal.

DOI: https://doi.org/10.7554/eLife.45636.007

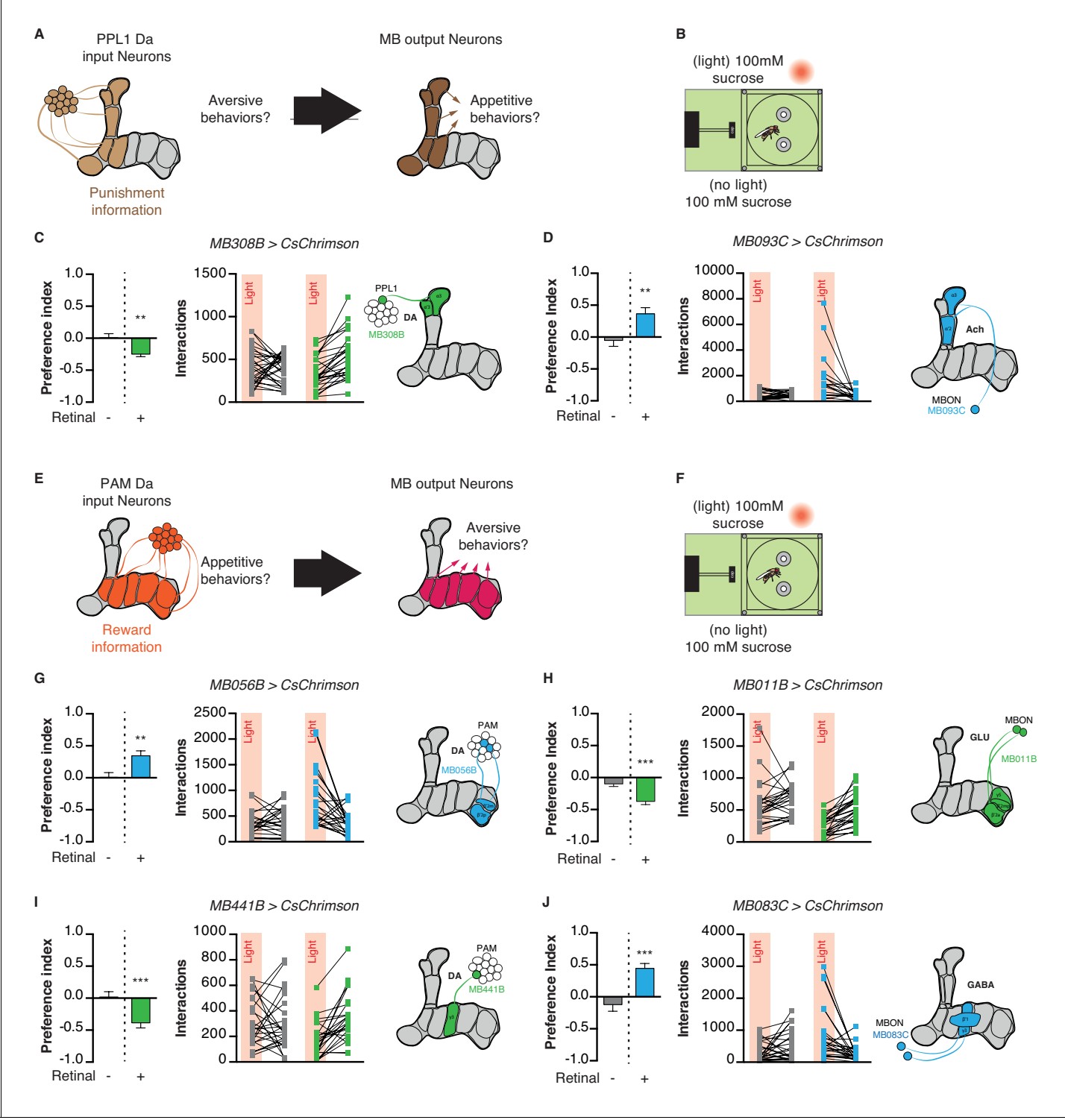

**Figure 6.** Manipulation of mushroom body extrinsic neurons modifies feeding behavior. (**A**) Model for PPL1 input to the MB and corresponding output: PPL1 neurons signal punishment and are predicted to drive aversive behavior, while corresponding MBONs are predicted to be appetitive. (**B**) Experimental setup: both channels contain 100 mM sucrose in 1% agar. (**C**) Light side preference and interactions of flies expressing CsChrimson in PPL1 neurons α3,α′3 (*MB308-GAL4*). (**D**) Light side preference and interactions of flies expressing CsChrimson in MBON α3,α′two neurons (*MB093C-GAL4*). (**E**) Model for PAM input to the MB and corresponding output: PAM neurons signal reward and are predicted to drive appetitive behavior while corresponding MBONs are predicted to be aversive. (**F**) Experimental setup: both channels contain 100 mM sucrose in 1% agar. (**G**) Light side preference and interactions of flies expressing CsChrimson in PAM β2,β′two neurons (*MB056B-GAL4*). (**H**) Light side preference and interactions of flies

*Figure 6 continued on next page*

*Figure 6 continued*

expressing CsChrimson in MBON neurons post-synaptic to PAM β2,β'two neurons (*MB011B-GAL4*). (I) Light side preference and interactions of flies expressing CsChrimson in PAM γ3 neurons (*MB441-GAL4*). (J) Light side preference and interactions of flies expressing CsChrimson in MBON neurons post-synaptic to PAM γ3 neurons (*MB083C-GAL4*). Values are mean ± SEM. n = 18–29. Statistical test: *t*-test. **p<0.01, ***p<0.001.

DOI: https://doi.org/10.7554/eLife.45636.027

The following source data and figure supplements are available for figure 6:

**Source data 1.** This file contains all the raw numerical data for *Figure 6* and its associated figure supplements.

DOI: https://doi.org/10.7554/eLife.45636.031

**Figure supplement 1.** Manipulation of PPL1 DANs and their corresponding output neurons modifies feeding behavior.

DOI: https://doi.org/10.7554/eLife.45636.028

**Figure supplement 2.** Manipulation of PAM DANs and their corresponding output neurons modifies feeding behavior.

DOI: https://doi.org/10.7554/eLife.45636.029

**Figure supplement 3.** Manipulation of PAM γ3 DANs and their corresponding output neurons modifies feeding behavior.

DOI: https://doi.org/10.7554/eLife.45636.030

Second, given the important role that PPL1s play in aversive conditioning, we examined whether activation of either population caused a shift in behavior over the course of the one-hour assay. However, as with GRN activation, relative interaction numbers remained quite stable throughout (*Figure 7—figure supplements 1* and *2*). Third, we tested whether silencing each population with expression of the inward rectifier potassium channel Kir2.1 would affect feeding behavior. After disabling LED activation in the STROBE, we counted flies' interactions with either a low concentration of sucrose or plain agar (*Figure 7—figure supplement 3*). This failed to reveal an effect of PPL1 or MBON silencing on food choice, indicating that, while activation of these populations is sufficient to affect food choice, they are not necessary in the particular task tested.

## Discussion

Leveraging real-time data from the FlyPAD, we built the STROBE to tightly couple LED lighting with sipping and other food interactions, thereby allowing us to optogenetically excite specific neurons during feeding. The primary advantage of the STROBE over existing systems for neural activation during fly feeding is its temporal resolution, which provides two key benefits. First, acutely activating neurons while the fly is choosing to interact with one of two available food sources allows us to explore the impact of neural activation on food selection in a way that is impossible with chronic activation. Second, by tightly coupling stimulation with food interaction events, light-driven activity from the STROBE should more closely mimic the temporal dynamics of taste input. Conceptually, these advantages are similar to those achieved by expression of the mammalian TRPV1 in taste sensory neurons and lacing food with capsaicin (*Caterina et al., 1997*; *Chen and Dahanukar, 2017*; *Marella et al., 2006*). Importantly, however, the STROBE allows activation of either peripheral or central neurons.

The implementation of interaction detection and light triggering on an FPGA allows the STROBE to trigger LED activation with minimal latency, well within the time frame of a single sip. Thus, neural excitation is tightly locked to the onset and offset of food interactions, providing the means to manipulate circuits during active feeding. Our decision to implement an algorithm that terminates illumination during capacitance plateaus exceeding 100 ms has both advantages and disadvantages. The main advantage is avoiding the LED becoming 'stuck' in the on state during shifts in baseline capacitance that could result in constant illumination for seconds or even minutes during which the fly may not be interacting with the food. A possible disadvantage is LED illumination that does not last the entirety of an interaction. This could produce insufficient activation to mimic specific properties of long sips. It could also result in suboptimal light-mediated silencing, although the efficacy of neural inhibition in the STROBE has not been tested.

These temporal qualities of light triggering are the primary difference between the STROBE and another recently described optogenetic FlyPAD, termed 'optoPAD' (*Moreira et al., 2019*; *Steck et al., 2018*). OptoPAD carries out sip detection and light control on a USB-connected computer. The benefits of this strategy are the ability to implement a more complex feeding detection algorithm, and more flexible control of the lighting response timing (*Moreira et al., 2019*). However,

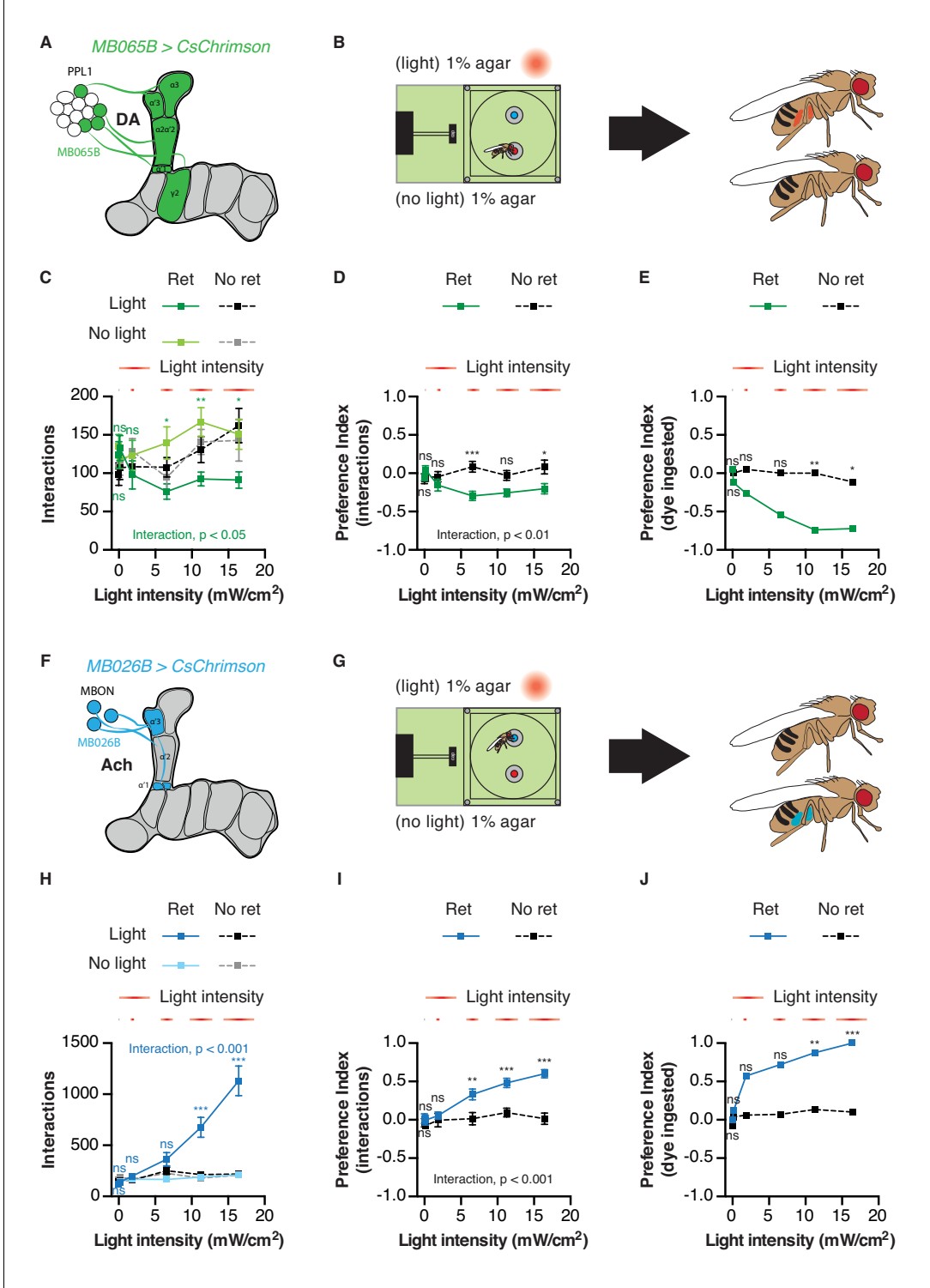

**Figure 7.** Activation of mushroom body circuits drives ingestion behavior. (A) Schematic of the PPL1 driver used. (B) Experimental setup: one channel contains 1% agar with blue dye, and the other has 1% agar with red dye. Fly abdomen color is determined following the experiment. (C) Food interactions from flies expressing CsChrimson in PPL1 DANs in the STROBE with increasing light intensity. Green lines indicate flies pre-fed retinal, and black/gray lines are non-retinal controls. (D–E) Light side preference indices calculated using interactions (D) or dye ingestion (E). (F) Schematic of the MBON driver used. (G) Experimental setup: one channel contains 1% agar with blue dye, and the other has 1% agar with red dye. Fly abdomen color is determined following the experiment. (H) Food interactions from flies expressing CsChrimson under control of *MB026B-GAL4* in the STROBE with increasing light intensity. Blue lines indicate flies pre-fed retinal, and black/gray lines indicate non-retinal controls. (I–J) Light side preference indices calculated using interactions (I) or dye ingestion (J). Values are mean ± SEM. n = 19–30 for (C–E) and 14–27 for (H–J). Statistical test: two-way ANOVA

*Figure 7 continued on next page*

*Figure 7 continued*

with Bonferroni post hoc. ns p>0.05, *p<0.05, **p<0.01, ***p<0.001. Colored asterisks represent significance between sips on each side for the retinal group.

DOI: https://doi.org/10.7554/eLife.45636.032

The following source data and figure supplements are available for figure 7:

**Source data 1.** This file contains all the raw numerical data for *Figure 7* and its associated figure supplements.
DOI: https://doi.org/10.7554/eLife.45636.036
**Figure supplement 1.** Behavioral dynamics during PPL1 DAN stimulation.
DOI: https://doi.org/10.7554/eLife.45636.033
**Figure supplement 2.** Behavioral dynamics during MBON stimulation.
DOI: https://doi.org/10.7554/eLife.45636.034
**Figure supplement 3.** Silencing PPL1 or MB026B neurons has no effect on sugar feeding.
DOI: https://doi.org/10.7554/eLife.45636.035

the tradeoff is longer and more variable latencies of LED activation. Moreover, the set illumination period further decouples the timing of illumination from the fly's behavior. Each of these systems may have specific advantages, depending on the application. While they have not been directly compared, it is likely that tight temporal coupling of the STROBE to food contact will be more useful for studying the effects of acutely activating core taste and feeding circuit neurons, while the longer, adjustable, light pulse from the optogenetic FlyPAD may be better for silencing neurons or activating reinforcement circuits.

Interestingly, similar closed-loop optogenetic paradigms have recently been developed for rodents. Lick-triggered blue light stimulation of taste receptor cells or circuits in the amygdala is sufficient to drive appetitive and aversive taste behaviors in mice (*Wang et al., 2018*; *Zocchi et al., 2017*). Thus, the same principle is able to reveal important insight into consummatory behaviors in multiple animals.

Although optogenetic neuronal activation is artificial, light-driven behavior in the STROBE shows some important properties that mimic natural feeding. For example, the number of food interactions evoked by stimulation of sweet sensory neurons increased in response to starvation, demonstrating a dependence on internal state that mirrors what is seen for sugar feeding (*Dus et al., 2013*; *Dus et al., 2011*; *Inagaki et al., 2014*; *Inagaki et al., 2012*; *Scheiner et al., 2004*). One curious observation is that the number of interactions driven by activation of sweet neurons peaks at a sub-maximal stimulus intensity, with fewer interactions observed at higher levels of illumination. Compensatory decreases in feeding are known to occur at high food concentrations, but this is likely from consumption of nutrients that are not present in the STROBE experiment (*Deshpande et al., 2014*). Since we observe no evidence of sensory-specific satiety over the time course of our experiments, we suspect that suppression of interactions may occur either from non-physiological neuron responses at high intensities or an interaction between visual cues from the increased light intensity and the sweet taste-mediated attraction. Regardless of the mechanism, the non-linear intensity-dependence of sweet neuron activation contrasts with that of bitter neurons and the MB neurons tested, which all show the largest behavioral responses at the highest light intensity. This demonstrates the important practical point that different neuron types can produce maximal effects at different stimulus intensities. Thus, it will be important for anyone using the STROBE or a similar optogenetic system to carefully calibrate the light intensity for their specific neurons of interest with the goal of matching either the neurons' physiological responses, or the animals' behavioral responses, to natural stimuli.

We also showed that the behavioral impact of sweet and bitter GRN activation in the STROBE could be abolished by the presence of natural taste ligands. Interestingly, this property did not generally hold true for attraction mediated by PAM or appetitive MBON activation, which was typically similar in the presence or absence of sugar. This may suggest that sweet taste input and PAM or MBON activation drive attraction via parallel circuits, producing an additive effect when both are present. Or perhaps suppression would be observed at higher sucrose concentrations or lower light intensities. It is also notable that flies preferred 1 M sucrose alone over 1 M sucrose coupled to optogenetic activation of sweet GRNs. We suspect that optogenetic activation of sweet GRNs in the STROBE plateaus below the excitation achieved with 1 M sucrose, and somehow prevents further

activation by very high sugar concentrations. We favor this interpretation over the alternative that the combination of 1 M sucrose with optogenetic activation of sweet neurons becomes 'too sweet'; however, both remain formally possible.

One interesting question is whether the valence of GRN activation in the STROBE is mediated by hedonics or effects on the feeding program itself. For example, sweet neuron activation is thought to carry appetitive hedonics, and therefore the flies may continue feeding because consequent light activation of Gr64f neurons is somehow pleasurable. On the other hand, these neurons also initiate feeding (and conversely, activation of Gr66a neurons terminates it). Thus, it is possible that each food contact evokes light-driven activation of a subsequent contact, and so on, creating a positive feedback loop. This is undoubtedly true of Fdg neuron activation, which is known to initiate a complete feeding sequence, likely downstream of any hedonic effects (*Flood et al., 2013*). Flies appear to become 'trapped' in a feeding loop until the end of the experiment, suggested by the very high number of evoked food interactions (see *Figure 5*).

Activation of MB input and output neurons also modulates feeding in the STROBE. PPL1 stimulation in the STROBE produces avoidance, while appetitive PAM stimulation produces attraction, consistent with the established valence of each population in memory formation and a previously reported role for PAMs in foraging behavior (*Aso et al., 2012*; *Burke et al., 2012*; *Aso et al., 2010*; *Das et al., 2014*; *Huetteroth et al., 2015*; *Kirkhart and Scott, 2015*; *Landayan et al., 2018*; *Lin et al., 2014*; *Liu et al., 2012*; *Masek et al., 2015*; *Yamagata et al., 2015*). Moreover, MBON activation drives feeding behavior in the opposite direction to activation of DANs from the same compartment. This relationship supports the current model that DAN activity depresses KC to MBON synapses in their respective compartments (*Cohn et al., 2015*; *Felsenberg et al., 2017*; *Perisse et al., 2016*; *Séjourné et al., 2011*; *Takemura et al., 2017*). Interestingly, not all PAM neurons convey a positive signal upon activation. PAM γ3 neurons are excited by electric shocks (*Cohn et al., 2015*) and inhibited by sucrose stimulation (*Cohn et al., 2015*; *Yamagata et al., 2016*). Our findings that PAM γ3 activation drives aversive feeding behavior is consistent with these neurons signaling negative valence to the MBs. Finally, it is worth noting that, although silencing experiments failed to reveal any requirement for PPL1s and their post-synaptic MBONs in the feeding paradigm used, it is possible that effects would be seen with silencing of broader neuron populations.

Although the mechanisms by which DANs affect feeding behavior remain unclear, MBONs can modulate innate behavior such as taste sensitivity (*Masek et al., 2015*), naïve response to odors (*Owald et al., 2015*), place preference (*Aso et al., 2014b*), and food seeking behavior (*Tsao et al., 2018*). Could the modulation of feeding by DAN activation result from learning? This seems unlikely with our current experimental design, as both food options were always identical, and thus there would be no predictive cues to associate with appetitive or aversive DAN stimulation. We think it is more likely that the same reward or punishment signals that underlie memory formation also acutely modify feeding behavior. However, the possibility of pairing circuit activation with specific food cues may offer a new paradigm for studying food memories, and neuronal activation via self-administration opens new avenues for the study of operant conditioning and addiction.

## Materials and methods

### STROBE system

The STROBE system consists of a field programmable gate array (FPGA) controller attached to a multiplexor board, adaptor boards, fly arenas equipped with capacitive sensors and lighting circuits. The hardware, with the exception of the lighting circuit units, is based on the FlyPAD design (*Itskov et al., 2014*). Each fly arena is paired with a lighting circuit and an opaque curtain (to prevent interference from external light). This pair will be referred to as a fly chamber unit. The entire system accommodates 16 fly chamber units (16 fly arenas and 16 lighting circuits), through eight adaptor boards. The FPGA used is a Terasic DEV0-Nano mounted onto a custom-made multiplexor board.

The multiplexor board is one of the intermediate connection components between the fly chambers and the FPGA controller. The multiplexor board has eight 10-pin ports each of which facilitate communications between two fly chambers and the FPGA controller. The board also has a FTDI module allowing data transfer over serial communications with a computer. The other intermediate connection component is the adaptor board which connects on one side to the multiplexor board

via a 10-pin line, and splits the 10-pin line from the multiplexor board into four 10-pin ports which connects to two fly arenas and two lighting circuits. The fly arena consists of two annulus shaped capacitive sensors and a Capacitance to Digital Converter (CDC) chip (AD7150BRMZ) that the main multiplexer board communicates with to initiate and collect data (and ultimately to stop collecting data). The CDC interprets and converts capacitance data from the two sensors on the fly-arena to a digital signal for the FPGA to process (*Itskov et al., 2014*).

The lighting circuit consists of a two-pin connector to receive power from an external power supply, a 10-pin connector to receive signals from the FPGA controller via the intermediate components, a 617 nm light emitting diode (LUXEON Rebel LED – 127lm @ 700mA; Luxeon Star LEDs #LXM2-PH01-0060), two power resistors (TE Connectivity Passive Product SMW24R7JT) for LED current protection, and two metal oxide semiconductor field effect transistors (MOSFETs; from Infineon Technologies, Neubiberg, Germany, IRLML0060TRPBF) allowing for voltage signal switching of the LEDs.

When a fly performs a sip and triggers a high signal on a capacitive sensor, the CDC chip on the fly arena propagates a signal via the multiplexor to the FPGA controller. The FPGA processes the capacitive sensor signal, decides a legitimate food interaction took place, and sends a high signal through the multiplexor to the MOSFET of the lighting circuit. The MOSFET then switches its lighting circuit on, allowing current to flow and turning on the monocolor LED positioned directly above the capacitive sensor. The process for determining a legitimate food interaction is described next.

In order to trigger optical stimulation with short latency upon food contact, we designed a running minima filter that operates in real-time to detect when a fly is feeding. We implemented this filter by modifying the state machine on the FPGA. When a fly feeds, its contact with the capacitance plate generates a 'step', or rising edge in the capacitance signal. Our filter determines the minimum signal value in the last 100 ms and checks whether the current signal value exceeds that minimum by a set threshold. This threshold (100 capacitance units) was selected to be large enough to discriminate rising edges of capacitance representing true food interactions from noise, but small enough to not miss true interactions. If this condition is true, the filter will prompt the lighting activation system to activate the LED (or keep it on if it is already on).

By design, this means that the control system will send a signal to deactivate the lighting upon the falling edge of the capacitance signal, or if the capacitance signal has plateaued for 100 ms, whichever comes sooner. At this point, a low signal is sent to the MOSFET which pinches off the current flowing through the lighting circuit, turning off the light. The signal to lighting response transition times are on the order of tens of milliseconds, providing a nearly instantaneous response.

After each lighting decision (on/off/no change), the system will then automatically record the state of the lighting activation system (on/off) and transmit this information through USB to the computer, where it is received and interpreted by a custom end-user program (built using Qt framework in C++) which can display and record both the activation state and signal measured by the STROBE system for each channel of every fly arena, in real-time.

All STROBE design materials are available as a supplemental download.

All STROBE software is available for download from Github:

FPGA code: https://github.com/rcwchan/STROBE-fpga (copy archived at https://github.com/elifesciences-publications/STROBE-fpga).

All other code: https://github.com/rcwchan/STROBE_software/ (copy archived at https://github.com/elifesciences-publications/STROBE_software).

## Latency measurements

A wire was attached to the outer electrode of the light-triggering channel, and agar was placed on the inner electrode as one would for a normal experiment. Video was captured at 178 frames/s, which corresponds to 5.6 ms per frame. Eight individual touches were analyzed by identifying the last frame where the wire was clearly not touching the agar. The subsequent frame was taken as the time of touch. The number of frames between the touch frame and the frame where LED activation is observed were then counted and multiplied by 5.6 ms to generate the latency of activation for that touch. We consider this a maximal estimate of latency, since the optics of the camera made close proximity and touch difficult to discriminate in some frames, in which case it was assumed that touch was occurring.

## Fly strains

Fly stocks were raised on standard food at 25°C and 60% relative humidity under a 12:12 hr light:
dark cycle. For neuronal activation experiments we used the *20XUAS-IVS-CsChrimson.mVenus* (in
attP40 insertion site) from the Bloomington *Drosophila* Stock Center (stock number: 55135). For neu-
ronal silencing experiments we used *UAS-Kir2.1* (*Baines et al., 2001*). Specific GRN expression was
driven using *Gr64f-GAL4* (*Dahanukar et al., 2007*) and *Gr66a-GAL4* (*Wang et al., 2004*).
*GMR81E10-Gal4* was used for expression in Fdg neurons (*Jenett et al., 2012*; *Pool et al., 2014*). All
MB split-GAL4 lines (*MB011B-GAL4; MB026B-GAL4; MB056B-GAL4; MB065B-GAL4; MB083C-
GAL4; MB093C-GAL4; MB10C-GAL4; MB195B-GAL4; MB210B-GAL4; MB308B-GAL4; MB310B-
GAL4; MB441B-GAL4*) were described in a previous study (*Aso et al., 2014b*) and obtained directly
from Janelia. The expression patterns of the lines from the Flylight collections are available from the
Flylight project websites.

## Fly preparation and STROBE experiments

All experiments were performed with female flies to reduce variability, given that sex differences
were not a subject of investigation. After eclosion, flies were kept for several days in fresh vials con-
taining standard medium, and were then transferred at 25°C into vials covered with aluminum foil
containing 1 ml standard medium (control flies) or 1 ml standard medium mixed with 1 mM of all-
*trans*-retinal (retinal flies) for 2 days. Flies were then subjected to a 24 hr fasting period where they
were transferred to covered vials containing 1 ml of 1% agar (control flies) or 1 ml of 1% agar mixed
with 1 mM of all-*trans*-retinal (retinal flies).

For the starvation curve experiment (*Figure 3*), flies were transferred into vials containing 1 ml of
standard medium ±all *trans*-retinal for 24 hr (fed group); or 1 ml of 1% agar ±all *trans*-retinal for 12-
24-48 hours.

All flies were 5–9 days old at the time of the assay, and experiments were performed between
10:00 am and 5:00 pm. Both channels of STROBE chambers were loaded with 4 µl of 1% agar with
or without sucrose (0, 1, 10, 100, 1000 mM) or denatonium (0, 0.1, 1, 10 mM). For aversive assays
using denatonium, 50 mM sucrose was also added to increase food interactions.

Acquisition on the STROBE software was started and then single flies were transferred into each
arena by mouth aspiration. Experiments were run for 60 min, and the preference index for each fly
was calculated as: (interactions with Food 1 – interactions with Food 2)/(interactions with Food
1 + interactions with from Food 2). The red LED is always associated to the left side (Food 1). For
temporal curves, data are pooled within 1 s time-period.

Sucrose, denatonium, agar and all-*trans*-retinal were obtained from Sigma-Aldrich.

For experiments done in *Figure 2*, light intensity used are 0, 0.12, 1.85, 6.56, 11.26 and 16.44
mW/cm$^2$. All the other experiments were performed with a light intensity of 11.2 mW/cm$^2$.

## Dye feeding assay

Both channels of the STROBE chambers were loaded with 4 µl of 1% agar, which contained either
0.125 mg/ml blue (Erioglaucine, FD and C Blue#1) or 0.5 mg/ml red (Amaranth, FD and C Red#2)
dye. Half the replicates for each experiment were done with the dyes swapped to control for any
dye preference. After the experiment, flies were frozen and scored for abdomen color. Preference
Index (PI) was scored as « 1 » for color associated to the light channel ; « −1 » for color associated
to the no-light channel ; and « 0 » for both colors.

## Immunohistochemistry

Brain immunofluorescence was carried out as described previously (*Chu et al., 2014*). Primary anti-
bodies used were chicken anti-GFP (1:1000, Abcam #13970) and mouse anti-brp (1:50, DSHB
#nc82). Secondary antibodies used were goat anti-chicken Alexa 488 (1:200, Abcam #150169) and
goat anti-mouse Alexa 568 (1:200, Thermo Fisher Scientific #A11004).

All images were acquired using a Leica SP5 II Confocal microscope with a 25x water immersion
objective. All images were taken sequentially with a z-stack step size at 2 µm, a line average of 2,
line-scanning speed of 200 Hz, and a resolution of 1024 × 1024 pixels.

## Statistical analysis

Statistical tests were performed using GraphPad Prism six software. Descriptions and results of each test are provided in the figure legends. Sample sizes are indicated in the figure legends.

Sample sizes were determined prior to experimentation based on the variance and effect sizes seen in prior experiments of similar types. All experimental conditions were run in parallel and therefore have the same or similar sample sizes. All replicates were biological replicates using different individual flies. Data for behavioral experiments were performed with flies from at least two independent crosses. There was one condition where data were excluded, which were determined prior to experimentation and applied uniformly throughout: the data from individual flies were removed if the fly did not pass a set minimum threshold of interactions (15), or the data showed hallmarks of a technical malfunction (rare).

## Acknowledgements

We thank Linda Au carrying out part of a screen of MBON activation in the STROBE, and Bonnie Chu for the fly drawing used in *Figure 2*. We also thank Carlos Ribeiro and Pavel Itskov for kindly providing their original FlyPAD VHDL code, which was instrumental in developing the STROBE system. This work was funded by Natural Sciences and Engineering Research Council (NSERC) grants RGPIN-2016–03857 and RGPAS 492846–16. MDG is a CIHR New Investigator and a Michael Smith Foundation for Health Research Scholar.

## Additional information

### Funding

| Funder | Grant reference number | Author |
| --- | --- | --- |
| Natural Sciences and Engineering Research Council of Canada | RGPIN-2016-03857 | Michael D Gordon |
| Natural Sciences and Engineering Research Council of Canada | RGPAS-49246-16 | Michael D Gordon |

The funders had no role in study design, data collection and interpretation, or the decision to submit the work for publication.

### Author contributions

Pierre-Yves Musso, Conceptualization, Formal analysis, Investigation, Visualization, Writing—original draft, Writing—review and editing; Pierre Junca, Formal analysis, Investigation, Visualization, Writing—review and editing; Meghan Jelen, Damian Feldman-Kiss, Investigation, Writing—review and editing; Han Zhang, Methodology, Writing—review and editing, Designed and built STROBE system with conceptual input from PYM and MDG, Wrote first draft of STROBE methods section; Rachel CW Chan, Software, Methodology, Writing—review and editing, Designed and built STROBE system with conceptual input from PYM and MDG, Wrote first draft of STROBE methods section; Michael D Gordon, Conceptualization, Supervision, Writing—original draft, Project administration, Writing—review and editing

### Author ORCIDs

Rachel CW Chan https://orcid.org/0000-0003-1009-6379
Michael D Gordon https://orcid.org/0000-0002-5440-986X

### Decision letter and Author response

Decision letter https://doi.org/10.7554/eLife.45636.041
Author response https://doi.org/10.7554/eLife.45636.042

## Additional files

### Supplementary files

• Supplementary file 1. This is a zip file containing all the information needed to build and operate a STROBE system.

DOI: https://doi.org/10.7554/eLife.45636.037

• Transparent reporting form

DOI: https://doi.org/10.7554/eLife.45636.038

### Data availability

All raw data are included as supplementary downloads.

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
