## [Decision Letter]

Thank you for submitting your article "Closed-looped optogenetic activation of peripheral and central neurons modulates feeding in freely moving *Drosophila*" for consideration by *eLife*. Your article has been reviewed by two peer reviewers, and the evaluation has been overseen by Mani Ramaswami as Reviewing Editor and Catherine Dulac as the Senior Editor. The reviewers have opted to remain anonymous.

The reviewers have discussed the reviews with one another and the Reviewing Editor has drafted this decision to help you prepare a revised submission.

Summary:

This manuscript describes the technical features and utility of STROBE, a closed-loop stimulation system, which improves on a version of FlyPAD described by Steck et al. (2018). STROBE allows light delivery – and therefore optogenetic activation of neurons – to be temporally coupled to flies' physical interaction with agarose substrates and has good throughput. The new system has tighter coupling of the onset of the light stimulus with feeding initiation, and so may claim greater relevance to the activity of neurons directly activated by the act of feeding or tasting. By quantifying how frequently flies triggered the light pulses, one can quantify how activation of specific groups of CsChrimson-expressing neurons modifies flies' interactions with specific agar substrates. The system will be useful for the field of feeding behavior, as studying how neuronal activation can change due to taste and how it can control behavioral outputs is critical to our understanding of neural circuits and maladaptive feeding relating to human disease. The manuscript's findings that the STROBE is an important new tool for acutely probing feeding behavior questions with manipulation of peripheral taste and central associative neurons is reasonably well supported by a few important proof-of-principle experiments with STROBE such as showing that flies with CsChrimson expressed in their sweet neurons significantly increase their interaction with a sugarless agar that was coupled to light. In addition, they also use STROBE to provide some new biology by demonstrating a potential role of specific dopaminergic neurons (DANs) and mushroom body output neurons (MBONs) in regulating feeding. This is interesting because these specific DA neurons are generally known for mediating reinforcement signals (US) during learning as opposed to regulating innate behaviors.

In general, the technique appears useful but the impact of the manuscript would be improved by both technical clarifications and by some experiments using it to provide increased biological insight.

Essential revisions:

1) The authors assess the accuracy of STROBE – i.e., whether LEDs are turned on when and only when flies sip – by comparing STROBE LED activations to sips detected by FlyPAD. While an independent analysis would be better, this seems acceptable assuming the sip detection in FlyPAD was done well. It is possible however, that for the main comparison between STROBE and FlyPAD (Figure 1E), bins with no sips called by either algorithm were excluded from analysis (according to the figure legend). This may hide errors STROBE may make compared to FlyPAD and so needs to be analyzed in more detail.

2) According to the algorithm shown in Figure 1C and its description in the sixth paragraph of the subsection “STROBE System”, STROBE is designed to turn off the LED after just 100 ms during longer sips. This is also shown in Figure 1D at the end of the trace, where FlyPAD detects a long sip for which only a fraction is illuminated by STROBE. While the manuscript claims this is "by design," a discussion of the impact on the experiments is required. A related issue is that in some figures the y axis was labeled "sips" and in some "interactions." Are the two terms are used interchangeably? The writing should be clarified here.

3) While it is interesting to see that optogenetic activation of punishment-mediating and reward-mediating DANs or specific MBONs impacted fly-substrate interactions in a manner that is consistent with known properties of these neurons, two additional experiments are needed to ascertain that these neurons indeed play a role in regulating feeding. First, one should probably directly quantify food consumption triggered by STROBE activation of these neurons. Second, one should silence these neurons and assess the impact of such silencing on food consumption. (Silencing experiments need not be done with STROBE of course). This is because activation of these neurons may not affect feeding per se; rather, flies may simply seek to increase or decrease activation of these neurons by staying around the light-coupled agarose or moving away from it (as suggested by Aso et al.).

4) Why was only one intensity used to stimulate neurons following the initial studies of intensity-dependent GRN activation? Sweet GRNs, as appetitive neurons, have a U-shaped (bi-directional) response curve to increasing intensity stimuli (Figure 2D, E), while aversive neurons (bitter GRNs) have a linear negative correlation with intensity (Figure 2I, J). This clearly indicates a fundamental difference in how appetitive and aversive stimuli are processed – and may indicate that sweetness intensities above a certain concentration may have aversive side effects, so it would be more appropriate to choose different stimulus intensities for appetitive vs. aversive stimuli, instead of using one stimulus for both. Furthermore, greater intensities may affect actual neuronal function in different ways, depending on the neuronal population (or the circuits they engage). Given the absence of direct measurements of light-evoked neuronal activity and since central-brain neuron populations likely have their own response dynamics distinct from those of the peripheral systems, it seem crucial to test different intensities of light stimulation for these circuits. In the absence of this experiment, even if the behavior moves in the expected directions, there is no reason to believe that the conditions used reflect a biologically relevant stimulus (i.e., consider different patterns of activation, such as slow tonic firing, or bursting, or no APs at all but only gradual changes in membrane potential.) As the intensity chosen was higher than the "optimal" intensity for sweet-GRN responses (and so produced a lesser preference despite greater stimulation), could a lower intensity have had a different effect in Figure 4B? Particularly, since many in the field may base their stimulation protocols on this manuscript after publication, a more detailed and careful evaluation of stimulus intensities for the DA+ neurons and MBONs, together with a discussion of the limitations of this new analysis, will be very useful.

5) The interpretation that increasing concentrations of sucrose to both wells induces an enhancement of aversion (see text for Figure 4C) for flies with bitter-GRN activation is not entirely convincing. An alternative explanation is that the flies become more attracted to the other well – this might be true especially if the light stimulus "traps" a neuron in a particular firing rate regardless of additional stimuli. Increasing only the sucrose concentration in the light-coupled well would clarify this point (as would some analysis of the neuron activation due to the light stimulus if this were possible.)

6) All experiments were done within an hour, and preference was described for the full hour. However, pairing activation only of sweet taste neurons with a starved fly's interaction with a non-caloric substance like 1% agar likely only has very acute effects. How does preference change over the course of the hour? Does the fly learn that there is no energy associated with the "sweet taste" of the light-coupled agar? If so, is this abolished with the DAN/MBON manipulations, or does it persist? A more in-depth temporal analysis of the feeding behaviors would improve the manuscript, especially since the FlyPAD records all the data.

[Editors' note: further revisions were requested prior to acceptance, as described below.]

Thank you for resubmitting your work entitled "Closed-loop optogenetic activation of peripheral or central neurons modulates feeding in freely moving *Drosophila*" for further consideration at *eLife*. Your revised article has been favorably evaluated by Catherine Dulac as the Senior Editor, Mani Ramaswami as the Reviewing Editor and two reviewers.

The reviewers acknowledge that the manuscript has been improved by extensive revisions. Chiefly among these are: a dyed-food consumption experiment which linked actual food intake with STROBE-measured interactions, and an experiment where the PPL1 neurons were deactivated using Kir2.1, to check their necessity in inducing feeding, and not simply light activation.

1) However, some recommended revisions were not addressed in the new manuscript: in particular to point #4 which was meant to guide the experiments toward more physiologically relevant neuronal firing, and thus more interpretable results. The authors performed a curve with PPL1 and MB026B but not with Gr64f. Addressing the reasons behind the U-shaped curve would have been more informative than the experiments conducted. If the authors deem is sufficient to carry out the experiments with the 6.5 mW/cm2 regardless of the GAL4 used, in the least they should provide a paragraph discussing the possibility that this stimulation isn't physiologically significant for the circuits they test (especially for those where ephys data are available) and that in any study using the STROBE, OPTOPAD or OPTOFLIC, all closed loop systems, should first calculate their own light intensity curve to more faithfully mimic a physiological relevant stimulation and outcome. Given that this will be an important resource for those working with sensory and reward circuits, addressing this point is paramount and the authors should encourage the readers to optimize the intensity to match the neurons' fire rate and to use 6.5 as a starting rather than finishing point.

---

## [Author Response]

Essential revisions:1) The authors assess the accuracy of STROBE – i.e., whether LEDs are turned on when and only when flies sip – by comparing STROBE LED activations to sips detected by FlyPAD. While an independent analysis would be better, this seems acceptable assuming the sip detection in FlyPAD was done well. It is possible however, that for the main comparison between STROBE and FlyPAD (Figure 1E), bins with no sips called by either algorithm were excluded from analysis (according to the figure legend). This may hide errors STROBE may make compared to FlyPAD and so needs to be analyzed in more detail.

We apologize for our ambiguous and possibly misleading wording. We excluded bins only if there were 0 sips/interactions called by *both* algorithms. These bins were interpreted as time periods when the fly was not interacting with the food on that channel, and therefore were not informative in determining the relationship between sip numbers called by each algorithm. The large number of these bins would have artificially increased the R-squared value, and thus overestimated the correlation between the algorithms. We have corrected the figure legend wording for clarity.

2) According to the algorithm shown in Figure 1C and its description in the sixth paragraph of the subsection “STROBE System”, STROBE is designed to turn off the LED after just 100 ms during longer sips. This is also shown in Figure 1D at the end of the trace, where FlyPAD detects a long sip for which only a fraction is illuminated by STROBE. While the manuscript claims this is "by design," a discussion of the impact on the experiments is required. A related issue is that in some figures the y axis was labeled "sips" and in some "interactions." Are the two terms are used interchangeably? The writing should be clarified here.

We have now described this property of the STROBE design more explicitly in the Results section, and added a passage on possible implications to the Discussion. We have also corrected all the figures to use the term “interactions” when referring to data from the STROBE. This was our intention in the original submission, but an old version of the figures was used in error.

3) While it is interesting to see that optogenetic activation of punishment-mediating and reward-mediating DANs or specific MBONs impacted fly-substrate interactions in a manner that is consistent with known properties of these neurons, two additional experiments are needed to ascertain that these neurons indeed play a role in regulating feeding. First, one should probably directly quantify food consumption triggered by STROBE activation of these neurons. Second, one should silence these neurons and assess the impact of such silencing on food consumption. (Silencing experiments need not be done with STROBE of course). This is because activation of these neurons may not affect feeding per se; rather, flies may simply seek to increase or decrease activation of these neurons by staying around the light-coupled agarose or moving away from it (as suggested by Aso et al.).

We have performed these experiments. First, we validated a dye-based consumption assay in the STROBE using GRN activation and saw that sweet or bitter neuron activation had the expected effects on consumption (Figure 2—figure supplement 4). We then used this assay to show that activation of PPL1 or its postsynaptic MBONs also affect consumption as predicted by the interaction numbers measured by the STROBE (Figure 7). In fact, we saw larger effects on our consumption metric than on interaction numbers in all cases.

We also performed silencing experiments on the same pair of PPL1 and MBON drivers. In this case we expressed Kir2.1 in each neuron set and measured interaction numbers and dye ingestion in the STROBE with no light stimulation (Figure 7—figure supplement 3). Flies were given a choice between 10 mM sucrose and plain agar. In this paradigm we saw no effects of PPL1 or MBON silencing on feeding, suggesting that activation of these circuits is sufficient to drive changes in feeding, but not necessary for a simple food choice.

4) Why was only one intensity used to stimulate neurons following the initial studies of intensity-dependent GRN activation? Sweet GRNs, as appetitive neurons, have a U-shaped (bi-directional) response curve to increasing intensity stimuli (Figure 2D, E), while aversive neurons (bitter GRNs) have a linear negative correlation with intensity (Figure 2I, J). This clearly indicates a fundamental difference in how appetitive and aversive stimuli are processed – and may indicate that sweetness intensities above a certain concentration may have aversive side effects, so it would be more appropriate to choose different stimulus intensities for appetitive vs. aversive stimuli, instead of using one stimulus for both. Furthermore, greater intensities may affect actual neuronal function in different ways, depending on the neuronal population (or the circuits they engage). Given the absence of direct measurements of light-evoked neuronal activity and since central-brain neuron populations likely have their own response dynamics distinct from those of the peripheral systems, it seem crucial to test different intensities of light stimulation for these circuits. In the absence of this experiment, even if the behavior moves in the expected directions, there is no reason to believe that the conditions used reflect a biologically relevant stimulus (i.e., consider different patterns of activation, such as slow tonic firing, or bursting, or no APs at all but only gradual changes in membrane potential.) As the intensity chosen was higher than the "optimal" intensity for sweet-GRN responses (and so produced a lesser preference despite greater stimulation), could a lower intensity have had a different effect in Figure 4B? Particularly, since many in the field may base their stimulation protocols on this manuscript after publication, a more detailed and careful evaluation of stimulus intensities for the DA+ neurons and MBONs, together with a discussion of the limitations of this new analysis, will be very useful.

We agree that manipulating stimulus parameters in different experiments could generate interesting results, in particular for the higher-order circuits in the mushroom body. However, we are also subject to practical limitations and must consider the cost-benefit ratio for additional experiments – for example, performing an experiment on one genotype under one condition across five light intensities requires about 25 2-hour (16 fly) experiments. Thus, performing intensity curves across five different conditions (e.g. different sucrose intensities in Figure 4B) would take 125 experiments, which would be well over a month of full-time work from one person. Given that the most likely outcome is that the curve simply shifts slightly in either direction by changing the intensities, this seems impractical.

However, we did perform intensity curves on two example mushroom body populations – the PPL1s and their postsynaptic MBONs (*MB026B-Gal4*). This data is now shown in Figure 7, which demonstrates that, in both cases, the effect size increases with increasing light intensities. Interestingly, unlike sweet GRN stimulation, which displays the noted U-shaped response, activation of these attractive MBONs produced linearly increasing interaction numbers across the range of intensities tested. Also, as we now point out in the text, we actually see a more noticeable effect on ingestion than we see on interaction numbers for both neuron populations tested, demonstrating that these circuits do indeed robustly affect feeding (as discussed above).

5) The interpretation that increasing concentrations of sucrose to both wells induces an enhancement of aversion (see text for Figure 4C) for flies with bitter-GRN activation is not entirely convincing. An alternative explanation is that the flies become more attracted to the other well – this might be true especially if the light stimulus "traps" a neuron in a particular firing rate regardless of additional stimuli. Increasing only the sucrose concentration in the light-coupled well would clarify this point (as would some analysis of the neuron activation due to the light stimulus if this were possible.)

We completely agree with this interpretation, and it was always our intention to suggest that it is the relative difference in attractiveness that is critical, and this is primarily driven by the increasing appetitiveness of the non-light side. We have reworded this passage to be more clear. Increasing the sugar concentration on the light-triggering channel only could show us the concentration of sucrose (if there is one) that is able to overcome the aversiveness of bitter GRN stimulation. However, given that light intensity is also a variable (and the experiment number limitations discussed above), this experiment is unlikely to reveal any fundamental insight into the system (or biology). We were more interested in examining the masking of GRN stimulation by taste ligands of the same modality.

6) All experiments were done within an hour, and preference was described for the full hour. However, pairing activation only of sweet taste neurons with a starved fly's interaction with a non-caloric substance like 1% agar likely only has very acute effects. How does preference change over the course of the hour? Does the fly learn that there is no energy associated with the "sweet taste" of the light-coupled agar? If so, is this abolished with the DAN/MBON manipulations, or does it persist? A more in-depth temporal analysis of the feeding behaviors would improve the manuscript, especially since the FlyPAD records all the data.

Detailed temporal data for GRN and DAN/MBON activation is now included in the supplements for Figures 2, 3, 4, and 7. As we discuss in the text, we generally see very stable relative interaction numbers (and therefore preference indices) after they are established in the first 15 minutes. This is true for peripheral and central neuron activation, and does not provide any evidence for sensitization, adaptation, or learning affecting the results we see over the course of a 1-hour experiment. This is not to say that learning does not (or cannot) occur in the STROBE, but this is an interesting avenue where, as we mention at the end of the Discussion, the STROBE could provide valuable insight in the future.

[Editors' note: further revisions were requested prior to acceptance, as described below.]

The reviewers acknowledge that the manuscript has been improved by extensive revisions. Chiefly among these are: a dyed-food consumption experiment which linked actual food intake with STROBE-measured interactions, and an experiment where the PPL1 neurons were deactivated using Kir2.1, to check their necessity in inducing feeding, and not simply light activation.1) However, some recommended revisions were not addressed in the new manuscript: in particular to point #4 which was meant to guide the experiments toward more physiologically relevant neuronal firing, and thus more interpretable results. The authors performed a curve with PPL1 and MB026B but not with Gr64f. Addressing the reasons behind the U-shaped curve would have been more informative than the experiments conducted. If the authors deem is sufficient to carry out the experiments with the 6.5 mW/cm2 regardless of the GAL4 used, in the least they should provide a paragraph discussing the possibility that this stimulation isn't physiologically significant for the circuits they test (especially for those where ephys data are available) and that in any study using the STROBE, OPTOPAD or OPTOFLIC, all closed loop systems, should first calculate their own light intensity curve to more faithfully mimic a physiological relevant stimulation and outcome. Given that this will be an important resource for those working with sensory and reward circuits, addressing this point is paramount and the authors should encourage the readers to optimize the intensity to match the neurons' fire rate and to use 6.5 as a starting rather than finishing point.

We agree that the U-shaped curve for Gr64f activation is notable and may reflect interesting biology that is worth pursuing. However, we also believe that truly understanding this phenomenon is not a straightforward prospect. The effect could be from non-physiological neuron responses at high intensities; a form of sensory-induced satiety; an interaction between the “virtual” sweet taste and visual responses to the flashing lights; or a host of other factors that are likely difficult to tease apart. The intention of this experiment was to illustrate that intensity was an important variable in the response, and the reviewer is correct that we could have highlighted this more clearly in discussing both the Gr64f and DAN/MBON intensity curves. We have now described the Gr64f result in more detail and added a passage to the Discussion (Discussion, fifth paragraph) that considers possible interpretations of the Gr64f result and emphasizes the general importance of calibrating light intensity to mimic physiological and/or behavioral responses to natural stimuli.